# Implementation of the Prevent-Teach-Reinforce Model for Elementary School Students Needing Intensive Behavior Intervention

**DOI:** 10.3390/bs14020093

**Published:** 2024-01-27

**Authors:** Sofia Ford, Kwang-Sun Cho Blair, Rose Iovannone, Daniel Kwak

**Affiliations:** 1Key Autism Services, Atlanta, GA 30303, USA; sofia.ford@keyautismservices.com; 2Department of Child and Family Studies, University of South Florida, Tampa, FL 33612, USA; iovannone@usf.edu; 3Kennedy Krieger Institute, Johns Hopkins University School of Medicine, Baltimore, MD 21205, USA; kwakd@kennedykrieger.org

**Keywords:** functional behavior assessment, individualized intervention, prevent-teach-reinforce, manualized intervention

## Abstract

This study evaluated the implementation of the school-based Prevent-Teach-Reinforce (PTR) model for elementary school students who engage in high levels of challenging behavior. Three students (one with speech or language impairment and two without disabilities) and their classroom teachers in two public schools participated in the team-based PTR process, which involved teaming and goal setting, functional behavior assessment, intervention, and evaluation. A multiple-baseline-across-participants design was used to evaluate the impact of PTR on student behaviors. Direct and indirect observations of student behaviors were conducted across target and generalization academic time periods. The findings indicated that the PTR intervention effectively improved the classroom behaviors of all three participating students in both target and generalization academic time periods, decreasing disruptive behavior and increasing on-task behavior. Social validity assessments with the participating teachers and one student indicated high levels of acceptability of and satisfaction with the PTR intervention goals, procedures, and outcomes.

## 1. Introduction

It is estimated that at least a third of all school-age children with or without disabilities will have an emotional or behavioral problem at some point during their school years [1]. A significant portion of these children may develop a chronic condition, eventually being diagnosed as having an emotional and behavioral disorder (EBD), such as conduct disorder and oppositional defiant disorder. Chronic externalizing behavior problems (e.g., physical aggression, property destruction, refusing to follow the rules) often lead to suspensions or expulsions, which impact the student’s academic performance [2]. Research indicates that students with EBD are more likely to experience long-term poor outcomes (e.g., dropping out of school, substance abuse, unemployment) compared to students without disabilities and those with other disabilities [3,4]. To mitigate these risks, efforts must be made to intervene the students engaging in significant behaviors of concern, who have not yet been identified as having an EBD, at an early age to help them succeed in school and in life.

Functional behavior assessments (FBAs) and function-based behavior intervention plans (BIPs) have been identified as effective practices for reducing behaviors of concern [5]. The purpose of conducting an FBA is to determine the behaviors of concern, the conditions under which the behaviors occur, and the environmental events that maintain these behaviors. The information gathered is then used to develop a BIP for the individual [6,7]. One factor that is essential in the efficacy of a BIP is a hypothesis regarding what the individual is accessing (or escaping) when engaging in the behavior, which is referred to as the function/s of behavior [8]. Aside from addressing the functions, all BIPs should involve procedures that are efficient and must consider the specific context in which they will be implemented [9]. Unfortunately, research shows that many of the FBAs and BIPs in school systems are inadequate, often not meaningfully involving teacher input, resulting in behavior plan strategies that are not implemented and are ineffective [10,11]. 

Research suggests that the most effective way to design and implement a behavior intervention plan (BIP) and reduce problem behavior while promoting appropriate behavior is through a team-based approach in functional behavior assessment (FBA) and BIP development and implementation [12,13]. Involving key school personnel, such as teachers, in the BIP development process is crucial for success. Benazzi and colleagues [14] found that, when a school-based team collaborated with someone knowledgeable about behavior analysis principles, the resulting plan was more likely to be technically sound. These team-developed plans also aligned with teachers’ values, skills, and practical implementation resources. Despite the research support for the team-based approach, there is a lack of standardized procedures available for school teams to utilize [15]. This results in the inconsistent application of FBA and BIP procedures which, in turn, impact the consistent and accurate implementation of a plan (implementation fidelity), which is a crucial determinant of its success [16,17]. 

Therefore, the use of intervention packages or manualized interventions has been explored in the literature to enhance efficiency, increase implementation fidelity, and optimize outcomes for students [18]. Smith [19] suggested that these manualized interventions should be replicable and adaptable to various individuals. Several manualized interventions designed for use by teachers or behavior support teams have been developed and assessed for their efficacy, with a few focusing on social competency [20,21]. However, only a limited number of these interventions address the functions of problem behavior [22,23]. One such manualized intervention addressing the functions of problem behavior and that has been tested in the school setting is the Prevent-Teach-Reinforce (PTR) model [24]. The PTR model offers a standardized approach, applying the FBA/BIP process in a school context. PTR follows a systematic five-step approach, including teaming, goal setting, assessment, intervention, and evaluation. All PTR BIPs include at least one intervention to prevent the behavior of concern from occurring, one to teach a replacement behavior, one to reinforce the replacement behavior with the function, and one to change how others (i.e., teacher) respond to the behavior of concern, ensuring that only the replacement behavior is reinforced according to the function. 

Several studies have shown the effectiveness of PTR. Iovannone et al. [24] used a randomized controlled trial to evaluate the PTR model with 245 students with disabilities or those at-risk for disabilities in grades K–8 and found it to be effective in decreasing student problem behavior and increasing academic engagement, as well as social skills. Dunlap et al. [25] illustrated two case studies on the use of PTR with typically developing children, which found significantly lower instances of problem behaviors (e.g., disruptive behavior, off-task behavior) and increases in prosocial behaviors (e.g., task engagement and following directions) during the PTR intervention implementation. DeJager and Filter [26] further examined the PTR process on three children without disabilities and found positive child outcomes. Strain and colleagues [27] assessed the PTR model on elementary school students with ASD in general education classrooms and found effective results. Similarly, Barnes et al. [28] demonstrated the positive outcomes of the PTR model when implemented with elementary school students at-risk for disabilities requiring individualized behavior interventions. The study showed improved behaviors for all three participating students. 

Although the current body of research on the school-based PTR model is promising, gaps do exist. Specifically, scant research explores whether teachers can implement PTR interventions in non-targeted routines and maintain student behavior change. Kulikowski et al. [29] came closest in their research that examined whether teachers in a community preschool classroom generalized PTR interventions to other children outside of the primary study participants. Using a concurrent multiple-baseline design across routines, two teachers (lead teacher and assistant teacher) implemented PTR with a targeted 4-year-old child, and the results indicated that the child’s behaviors improved after the PTR intervention. The teachers were then asked to develop and implement a PTR intervention with minimal researcher involvement for another child. Data indicated that the teachers were able to implement a PTR intervention with adequate fidelity for a non-target child. In addition, the child’s target behaviors showed improvement compared to the baseline. 

Therefore, the purposes of this study were to further evaluate the efficacy of PTR for elementary students (K–5) with an increased chance for EBD who engage in high levels of behaviors of concern during academic activities and to explore the generalization of PTR. This study addressed the following questions: (a) To what extent does PTR decrease disruptive behavior and increase replacement behavior in students with or without disabilities who have an increased chance of being identified with an EBD due to exhibiting high levels of chronic behavior problems?; (b) To what extent do teachers implement the PTR intervention plan with fidelity in academic activities other than those targeted for intervention and maintain child behavior changes?; and (c) Do teachers and students consider the PTR intervention goals, procedures, and outcomes acceptable?

## 2. Method

### 2.1. Setting and Participants

This study was conducted in three general education classrooms at two public elementary schools in a large city in a southern state. The PTR intervention was implemented during naturally occurring classroom activities, in which the participating students engaged in high levels of behavior that interfered with learning and positive social interactions. One school had approximately 1000 students, with 51% White, 26% Hispanic, 11%, Black, 9% Multi-Racial, and 3% Asian. The other school had approximately 700 students, with 58% White, 23% Hispanic, 8% Black, 7% Multi-Racial, 3.8% Asian, and 0.2% Native American. Both schools were implementing a multi-tiered system of support framework to address students’ academic, behavioral, and social–emotional needs across tiers of support.

Three students with or without disabilities who were engaging in high levels of intensive behavior that interfered with academic activities and who were receiving supplementary Tier 2 academic or behavioral support participated in the study. The inclusion criteria for student participants included: (a) they exhibited disruptive behavior (e.g., calling out, noncompliance) during classroom routines or activities for at least 3 out of 5 school days and for two or more behavioral incidents each day, (b) they were between the ages of 5 and 10, and (c) they had an attendance rate of 80% of school days or higher. Four teachers participated in the study. The inclusion criteria for teacher participants included: (a) regularly interacting with the student participant during classroom activities, (b) willing to participate in the PTR team process, and (c) willing to implement the PTR intervention during their classroom activities. Teachers who had prior experience with PTR were excluded from this study. Additional school personnel, a school psychologist and a guidance counselor who knew the student participant well, participated in the study as the students’ PTR team members. 

Following approval from the university and school district institutional review boards, flyers describing the study’s purpose, target population, and behaviors of interest were provided to K–5 teachers for participant recruitment. Four interested teachers met with the researcher (first author) to review informed consent forms. Teachers distributed flyers and parental permission forms to all classroom students in their classrooms. Upon receiving parental consent, the researcher obtained written or verbal assent from the three potential student participants. Teachers subsequently completed screening forms with open-ended questions about each consented student’s behavior. To confirm that all inclusion criteria were met, the researcher conducted two 30 min observations during problematic academic activities (e.g., math, independent work, transitions between activities) for each potential student participant. After completing the screening and observation procedures, all three students met the full eligibility criteria and were officially enrolled as participants in the study. 

#### 2.1.1. Student Participants

Ryan was an 8-year-old White, non-Hispanic boy enrolled in the 3rd grade. He was neurotypical without identified disabilities and exhibited behaviors of concern in and out of the classroom. His school day was split between two teachers, science/math and reading/writing. Ryan was receiving Tier 2 support for reading and writing. The behavior exhibited by Ryan was related to disrespect toward others, minimal academic engagement, and rarely completing assignments. Typical responses included reprimands and office discipline referrals (ODRs). The teachers were concerned that Ryan may repeat 3rd grade. The researcher conducted two 30 min observations during an academic routine identified as a time in which Ryan exhibited behaviors after the initial screening to confirm his eligibility. During the observations, Ryan exhibited disruptive behavior for most of the academic time period.

Pete was a 6-year-old White, non-Hispanic boy in the 1st grade with no identified disabilities. His behaviors of concern included leaving his area, throwing items, verbal disruptions, and screaming. Pete received Tier 2 behavioral support, consisting of a behavior chart with positive reinforcement and participation in a social skills group. During the current school year, Pete received 3 office discipline referrals (ODR), each resulting in out-of-school suspensions (OSS). Pete was performing below grade level in reading, writing, and math. The school suggested an FBA to better understand his behaviors. During the two initial 30 min observations, Pete exhibited behaviors consistently throughout both sessions, necessitating his removal from the classroom for a “cool down walk” on one occasion.

Toby was a 6-year-old White, non-Hispanic boy enrolled in the 1st grade who was on an Individualized Education Plan (IEP) for receiving special education services, falling under the disability category of speech or language impairment. His behaviors of concern included inappropriate interactions with peers, not keeping his hands to himself, and refusing to work on assignments. Responses to behaviors included withdrawing task demands and sending unfinished assignments home to complete as homework. During the two initial 30 min observations, the researcher observed Toby engaging in refusing to follow directions for most of the academic period on both days.

#### 2.1.2. Teacher Participants 

All four teachers were White, non-Hispanic women. Ryan’s primary teacher was 37 years old with a bachelor’s degree in Elementary Education and 13 years of teaching experience. She taught a third-grade class consisting of 18 students, including 4 with IEPs. The class had 12 male and 6 female students with 13 White, 3 Asian/Pacific Islander, 1 Hispanic, and 1 Multi-Racial students. Existing classroom management systems included verbal redirection, reprimands, time-outs, and a clipboard system aligned with classroom expectations and rules. Ryan’s secondary teacher was 35 years old with a bachelor’s degree in education and 9 years of teaching experience. She also taught the same third-grade classroom as Ryan’s primary teacher. 

Pete’s teacher was 48 years old with a master’s degree in educational leadership and 21 years of teaching experience. She taught a first-grade class consisting of 17 students, seven with IEPs. The class had 11 male and 6 female students, including 9 White, 2 Black, 5 Hispanic, and 1 Multi-Race students. Existing classroom management systems included the use of colored cards, where students would pull a red card after a verbal warning. Depending on the behavior, consequences included notifying the student’s parents, loss of recess time, or loss of classroom privileges. 

Toby’s teacher was 33 years old with a bachelor’s degree in education and 7 years of teaching experience. She taught a first-grade class with 16 students, including 3 with IEPs. The class had 9 male and 7 female students, with 9 White, 1 Black, and 6 Hispanic students. The existing classroom management system was a color clip chart, where students would move their clips up or down based on complying with or violations of classroom rules. Additional responses included verbal reprimands and redirections.

### 2.2. Measurement

Systematic direct observational data were collected throughout the study to evaluate the outcomes of the PTR intervention for participating students. Specifically, the observers (researcher and research assistant) used a continuous measurement procedure (i.e., duration) to measure targeted behaviors for reduction and replacement behaviors. The direct observational data were collected using a mobile application, Countee, that allows for collecting real-time data on continuous and discrete responses [30]. Additionally, teachers used the Individualized Behavior Rating Scale Tool (IBRST; [31]) to gather indirect observational data on student target behaviors of concern and replacement behaviors. Other supplementary data, such as treatment fidelity and social validity, were collected to further evaluate the PTR intervention process and outcomes.

#### 2.2.1. Direct Observation of Behavior of Concern and Replacement Behavior

Target behaviors of concern and replacement behaviors were identified and operationally defined individually for each participant. The researcher collaborated with each student’s PTR team to provide examples and non-examples, guiding the behavior definitions. Disruptive behaviors, engaged in by all three students and targeted for reduction, were operationally defined, with on-task behavior identified as the replacement behaviors to be increased. The length of direct observation remained consistently set at 30 min for all students and sessions. A 5 s duration criterion was used to define the topographies of both disruptive and on-task behaviors for all three students. Specifically, the behavior was required to persist at least 5 consecutive seconds to be included in duration measurements. The mobile data collection application, Countee, allowed for duration recording per occurrence and for the calculation of total duration across occurrences.

Ryan. Ryan’s team, including the two classroom teachers and the guidance counselor, operationally defined his disruptive behavior (behavior of concern) as engaging in any behavior other than the assigned task or ongoing activity, which included fiddling with items in his desk, doodling, putting his head down on the desk, and inappropriate use of classroom materials (e.g., tapping a pencil, writing on desk, folding assignments into origami). On-task behavior (replacement behavior) was defined as actively engaging in assigned tasks or ongoing activity, which included looking at the teacher during instruction and following teacher directives (e.g., put folder away, open textbook to assigned page, start independent work) while engaging in an instructional activity and using classroom materials as needed to complete assignments (e.g., paper and pencil to solve a math problem).

Pete. Pete’s team, including the classroom teacher and school psychologist, defined disruptive behavior as engaging in any behavior that did not match the delivered instruction or ongoing activity. This included leaving the assigned area without permission and making sounds that could be heard from at least 5 feet away (e.g., hitting or kicking table, talking to peers). On-task behavior was defined as engaging in assigned work or ongoing activity, remaining within the designated area, maintaining a quiet voice, and following classroom rules, such as raising a hand to speak, asking for help, or requesting permission. 

Toby. Toby’s team, including the classroom teacher and school psychologist, defined disruptive behavior as refusing to follow directions, engaging in verbal defiance (e.g., saying “no” to teacher directives), and engaging in inappropriate use of classroom materials (e.g., throwing or crumbling up worksheets). Toby’s team defined on-task behavior as engaging in an ongoing instructional activity. This included paying attention to teacher instructions, following directions while engaging in the given activity, and using classroom materials appropriately (e.g., using a writing utensil and paper to complete an assigned activity). 

#### 2.2.2. Individualized Behavior Rating Scale Tool (IBRST)

Teachers collected data on the participating students’ target behaviors using the IBRST [31], which was developed collaboratively by each student’s PTR team members, including the teachers. The IBRST is a direct behavior rating that combines direct observation with a rating scale with a 5-point Likert-type rating. For this scale, a rating of 5 indicates a high occurrence of the behavior (both behaviors of concern and replacement behaviors), whereas a rating of 1 indicates a low occurrence. Teachers rated each targeted behavior based on their perception of its occurrence. Ratings were based on each unique student’s presentation of the behaviors. For all three students, the IBRSTs used duration or percentage of time the student engaged in the disruptive behavior or the replacement behavior during a specified academic time period with high behavior rates. The teachers selected time periods for collecting baseline and intervention data as well as for generalization data. 

For both Ryan and Pete, math was selected as the time period for evaluating behavior change data. Generalization time periods selected were reading for Ryan and writing for Pete. Duration was chosen as the measurement basis. For both students’ disruptive behaviors, a rating of 5 was characterized by 25–30 min of engagement in the behavior. For replacement behavior, a rating of 5 was characterized as 25–30 min of engagement in replacement behavior. For Toby, the literacy blocked time was selected as the instructional period and science as the generalization period. Percentage of time was selected to represent his target behaviors. A rating of 5 for disruptive behavior was characterized by Toby engaging in disruptive behavior for 80–100% of the time period. A rating of 5 for replacement behavior was characterized by engaging in the replacement behavior for 80–100% of the time period. Teachers were instructed to complete the IBRST after the researcher conducted direct observation. 

#### 2.2.3. Integrity of PTR Process

The integrity of the PTR process was assessed using the PTR Integrity Checklist to assess whether the researcher followed each step of the process as intended during each meeting or step. The checklist included 54 items aligned with the 5 steps of the PTR process. The checklist consisted of 13 items for teaming and goal setting, 8 items for PTR assessment (FBA), 26 items for intervention and coaching, and 7 items for evaluation. The results indicated that the researcher implemented the PTR process with 100% integrity for all three teams, suggesting that the researcher consistently followed the intended steps of the process in each meeting.

#### 2.2.4. Teacher Implementation Fidelity

The observers (first author and research assistant) collected data on each teacher’s implementation of the PTR intervention plan using the PTR implementation fidelity checklist. The checklists were tailored to each individual student’s PTR intervention plan and were designed to assess whether the steps of each student-specific intervention plan (i.e., prevent, teach, and reinforce) were correctly implemented by the teacher. The fidelity checklist included 12 to 14 observable behaviors, with the number depending upon the specific strategies selected for the individual student. Implementation fidelity was quantified as a percentage by dividing the number of correctly implemented steps by the number of total steps, then multiplying the result by 100 for each session during the intervention phase of the study. 

Teacher implementation fidelity was assessed in 100% of observation sessions for all participants. These observations occurred during target and generalization time periods. Ryan’s teacher implemented the intervention plan, on average, with 86% fidelity (range, 62–93%) during the target instructional time period. During the generalization time, the secondary teacher implemented the plan, on average, with 95% fidelity (range, 92–100%). Pete’s teacher showed strong fidelity with an average rate of 99% (range, 90–100%) during the target time period and 100% fidelity during the generalization time. Toby’s teacher had an average fidelity rate of 93% (range, 91–100%) during the target instructional time and 92% fidelity (range, 82–100%) during the generalization time.

#### 2.2.5. Interobserver Agreement

The researcher and a trained research assistant collected data to assess the interobserver agreement (IOA) on student behaviors, researcher procedural integrity, and teacher implementation fidelity. The researcher and research assistant were both master’s students in Applied Behavior Analysis who had been providing consultation services to teachers across several schools, including the participants’ school, as their internship training. Both observers collected data simultaneously and independently during the direct observation sessions. IOA on student target behaviors was assessed in 37% of the sessions on average (range, 33–43%) across experimental phases for all students. IOA was calculated by dividing the shorter duration by the longer duration and multiplying by 100. The IOA for researcher procedural integrity of the PTR process implementation and teacher fidelity of intervention implementation was assessed in 40% of meetings or sessions across the target and generalization time periods (range, 36–43%). IOA was calculated by dividing the number of agreements by the total number of steps in the task analysis and multiplying by 100. 

For disruptive behavior, the mean IOA was 97% for Ryan, 97% for Pete, and 96% for Toby. For on-task behavior, the mean IOA was 97% for Ryan, 98% for Pete, and 96% for Toby. With the exception where Toby’s IOA was 83%, all IOAs ranged from 91% to 100%. During the baseline phase, IOA averaged 97% (range, 84–100%) across the target and generalization periods. In the intervention phase, IOA averaged 97% (range, 83–100%) across the target and generalization time periods. Both researcher procedural integrity and teacher implementation fidelity achieved 100% IOA in all assessed sessions.

#### 2.2.6. Social Validity

The PTR social validity measure was used with the participating teachers to assess their acceptability of the PTR intervention. The social validity evaluation tool was adapted from the Treatment Acceptability Rating Form–Revised (TARF-R; [32]), which consisted of 18 items and used a 5-point Likert-type scale. In addition, student participants were asked to complete a researcher-developed social validity questionnaire. This questionnaire consisted of 5 questions that were rated on a 5-point Likert-type scale. The purpose of these assessments was to gauge the acceptability of the PTR intervention from the perspective of both the teachers and the student participants.

### 2.3. Experimental Design

A multiple-baseline design across participants was used to demonstrate a functional relation between PTR and student behavior change. In this design, two students underwent concurrent baseline phases. Due to the pressing need for intervention with Ryan and a delay in securing parental consent (permission) for Pete and Toby, the baseline measurement for Ryan commenced 6 days earlier than those for the other two students. Pete and Toby underwent baseline measurement simultaneously on the same day. Five to nine baseline data points were gathered for each student before implementing the intervention. The introduction of intervention was staggered across students, which did not require withdrawing the intervention to demonstrate experimental control. In addition to visual analysis, Tau-BC [33] was calculated to determine the effect size of the outcomes. Tau-BC is an effect size that examines the overlap between the baseline and intervention phases when pairwise comparisons are made between those two conditions, and it also corrects for a baseline trend. Tau-BC was used as a secondary analysis, and decisions regarding the introduction of intervention for each participant were made based on visual analysis.

### 2.4. Procedures

The PTR process included five steps that were implemented through a series of team meetings throughout the study. The five steps were: (a) teaming, (b) goal setting, (c) PTR assessment, (d) PTR intervention and coaching, and (e) evaluation. The researcher served as the facilitator and guided each team through the process. Meeting times were approximately 30 min across meetings and teams. 

#### 2.4.1. Meeting 1: Steps 1 and 2—Teaming, Goal Setting, and IBRST Development

Ryan’s PTR team included his two teachers and his mentor, the school guidance counselor, who checked in with him once a week. Pete’s team consisted of his teacher and the school psychologist, who had worked closely with him throughout the year. Toby’s team comprised his teacher and the school psychologist, who interacted with the student on a daily basis at school. The school psychologist was on both Pete and Toby’s team. Once teams were established, each member identified behaviors they wished to see each student decrease and then increase. A consensus was reached in each team on the target behaviors to be selected for the FBA and the operational definitions. Following this, the IBRST was developed (as described earlier). At the conclusion of the meeting, the researcher gave each team member a PTR assessment and asked them to complete the assessment and return it to the researcher before the next meeting. The PTR assessment used a checklist format, adapted from structured FBA interviews, which was structured into three sections. The prevent section inquired about antecedent events that occasioned both concerning behaviors and engaged behaviors. The teach section asked team members to identify the suspected functions of the concerning behavior. The reinforce section asked about consequences following concerning behavior and asked about additional reinforcement. The PTR assessment was used as an interview, a checklist, or a combination, depending on the preferences of the team. Before the next meeting, the researcher compiled the findings from two observations that were conducted using an ABC narrative recording during the initial screening phase. These descriptive FBA data were combined with team member responses to the PTR assessments to develop a draft hypothesis statement for review, clarification, and consensus. 

#### 2.4.2. Baseline Data Collection

After the first meeting, the teachers started collecting baseline data on each student’s target behaviors using the IBRST. The observers (i.e., researcher and research assistant) also collected baseline data on the student’s target behavior using a continuous measurement system (i.e., duration). During the baseline, teachers responded to student behavior using their existing classroom management systems, as they normally would. The observers did not interfere with typical classroom activities during direct observations. Baseline data were collected for a minimum of five sessions and until the data were stable or demonstrated a trend toward the counter-therapeutic direction.

#### 2.4.3. Meeting 2: Step 3—PTR Assessment

The second team meeting reviewed the draft hypotheses. Clarification about the antecedents, functions, and consequences of the disruptive behavior was obtained from team members until a consensus was reached on the final hypothesis. The final hypotheses for each student’s disruptive behavior were as follows:

Ryan. When given a directive to perform a non-preferred task, specifically independent or small group work in math, reading, or writing, he would engage in disruptive behavior, including fiddling with items in is desk, doodling, putting his head down on his desk, and inappropriately using classroom materials. As a result, he would terminate or delay the non-preferred task.

Pete. When prompted to engage in a non-preferred activity (e.g., writing and math) while the teacher was attending to another student, he would engage in disruptive behavior, including leaving his assigned area without permission and making sounds that could be heard from at least 5 feet away by hitting or kicking the table or talking to peers for more than 5 s. As a result, he would (a) gain adult attention and (b) terminate or delay the non-preferred activity.

Toby. When prompted to complete independent seatwork requiring writing, he would engage in disruptive behavior including refusing to follow directions, engaging in verbal defiance such as saying “no” to teacher directives, and engaging in inappropriate use of classroom materials, such as throwing or crumbling up worksheets.

#### 2.4.4. Meeting 3: Step 4—PTR Intervention and Coaching

Using a PTR intervention menu that included evidence-based behavior strategies, each team member ranked two to four interventions from the three categories, prevent (antecedent strategies), teach (replacement behavior strategies), and reinforce (changing contingencies following replacement and concerning behaviors). The researcher provided guidance to team members on the strategies that best matched each student’s hypothesis, providing descriptions and examples. The highest-ranking strategies best matching the hypotheses in each category were selected for each student’s intervention plan. The researcher collaborated with each teacher to develop a task analysis of the selected strategies that would be feasible for implementation in the classroom. Table 1 displays summaries of the strategies selected for each student’s behavior plan. 

As presented in Table 1, all three teachers selected providing choices as a prevent strategy, although the choice options were different. Toby’s teacher also selected classroom management due to the desire to address behaviors of all students. On-task was the replacement behavior selected by all three teachers. Each teacher selected an additional behavior to teach, including problem-solving (Ryan), general coping strategies (Pete), and social skills (Toby). Reinforce strategies included providing the functional reinforcers (escape or attention) when each student met their on-task behavioral goals. Pete’s teacher also delivered behavior-specific praise as a way to give attention as a reward for on-task behavior continuously throughout a task. All three teachers selected redirecting students to their replacement behaviors if a student engaged in the concerning behavior.

PTR Intervention Plan Training. The researcher used behavioral skills training (BST; [34]) to train the teachers on implementing the plan before beginning the intervention. BST procedures included the researcher explaining the plan and modeling the steps. The teacher and the researcher then role-played the plan, with the researcher providing the teacher performance feedback. Training sessions continued until the teachers were able to role-play the plan with 90% fidelity. Both of Ryan’s teachers were trained on his plan and had an opportunity to practice at least once, which took approximately 20 min. Pete and Toby’s teachers were trained individually, and each took approximately 15 min. After the training, Ryan’s teachers separately met with him for approximately 5 min in class to review the intervention strategies. Upon the teacher’s request, the researcher met with Pete for about 10 min of non-educational time in the school psychologist’s office to review his strategies with him. Toby’s teacher met with him for about 5 min of non-educational time at the beginning of the day to review his plan.

PTR Intervention Plan Implementation. Following the training, the teachers implemented the PTR intervention plans within a day during the target instruction periods. The researcher briefly met with each teacher for about 5 min daily for the first few implementation days to provide coaching, based on observations of student behavior and implementation fidelity measurement. Each coaching session included (a) identifying areas of success, (b) asking the teacher for reflection on parts of the plan that were difficult or challenging, (c) providing suggestions for addressing challenging parts of the plan, (d) answering questions, and (e) ending the session with a positive statement related to teacher performance. For any teacher whose implementation fidelity score fell below 80%, a booster training session would be provided by the researcher or, if the teacher expressed feasibility challenges with the intervention plan, modifications would be made. Only Ryan’s primary teacher needed a booster training on the second day of intervention when her implementation fidelity fell below 80%. Following the booster training, the teacher’s fidelity remained above 80%.

Each teacher selected another part of the school day when concerning behaviors were present to implement the intervention for generalization probes. Ryan’s secondary teacher elected to implement the plan during reading, Pete’s generalization period was writing, and Toby’s teacher selected science as the target generalization time period. The researcher did not provide any additional training to the teachers to implement the plan during the non-target generalization time period; however, feedback was provided at the end of the initial generalization session.

#### 2.4.5. Meeting 4: Step 5—Progress Monitoring and Data-Based Decision Making

To address Step 5 of the PTR process, the researcher monitored student progress and teacher implementation fidelity throughout the intervention phase, while teachers continued daily use of the IBRST to rate student behaviors. Each team held a 30 min fourth meeting to review each student’s progress and teacher implementation fidelity and make data-based decisions, including modifying intervention procedures if necessary and promoting generalization and maintenance effects over time. However, teams were content with the students’ progress overall and focused the discussion on strategies to ensure continued improvements and maintenance of student behaviors upon study completion. At the end of the meeting, the teachers anonymously completed the TARF-R social validity measure in a sealed envelope to evaluate the acceptability of the PTR intervention.

#### 2.4.6. Follow-Up

Three or four follow-up probes were conducted for Ryan and Pete during both the target and generalization academic time periods, approximately 2–6 weeks after the last day of the intervention phase. The purpose was to assess whether the teachers continued to implement the PTR intervention without researcher involvement and whether treatment effects had been maintained.

## 3. Results

### 3.1. Student Behavior

#### 3.1.1. Direct Observational Data

Figure 1 displays data collected through direct observation of the three students’ concerning and replacement behaviors. For all three students, disruptive behavior decreased and replacement behavior increased upon introduction of the intervention. During baseline, Ryan engaged in disruptive behavior for an average of 20.1 min (range, 12.3–25.5 min) and engaged in on-task behavior for an average of 9.3 min (range, 8–16.9 min) during the target instructional period (i.e., math). During baseline, Ryan’s disruptive behavior remained relatively high, with the exception of the third day, during which he engaged in on-task behavior for a longer period of time than his engagement in disruptive behavior. During intervention, Ryan’s disruptive behavior decreased to an average of 4.5 min (range, 1.7–8.7 min), and his on-task behavior increased to an average of 24.3 min (range, 20.7–27.8 min). Ryan’s intervention data demonstrated a reduction in disruptive behavior and an increase in replacement behavior immediately after the intervention was implemented. Additionally, there were no overlapping data points between baseline and intervention phases for both disruptive behavior and on-task behavior. Ryan demonstrated stable intervention data with little variability. At the end of the intervention, Ryan exhibited an increasing trend for on-task behavior and a decreasing trend for disruptive behavior. During follow-up, Ryan engaged in disruptive behavior for 6.2 min and on-task behavior for 23.9 min. 

Pete engaged in disruptive behavior for an average of 23.5 min (range,17.6–29.6 min) and engaged in on-task behavior for an average of 4.6 min (range, 0.3–6.8 min) during baseline. Pete’s disruptive behavior increased after the first day of baseline and gradually decreased before increasing again and becoming more stable. During the intervention, Pete’s disruptive behavior decreased to an average of 1.2 min (range, 0–3.3 min), and his on-task behavior increased to an average of 28.3 min (range, 26.5–29.7 min). Upon introduction of the intervention, Pete’s data indicated an immediate reduction in disruptive behavior and an increase in replacement behavior. Pete demonstrated stable intervention data with little variability, and there were no overlapping data points between the baseline and intervention phases. During the follow-up period, Pete demonstrated maintenance of improvement in his reduced disruptive behavior, with an average duration of 2.7 min (range, 1.0–2.8 min), and an increase in on-task behavior, with an average duration of 26.3 min (range, 24.8–28.9 min).

Toby engaged in disruptive behavior for an average of 25.6 min (range, 22.2–29.3 min) and engaged in on-task behavior for an average of 3.8 min (range, 0.4–5.9 min) during the baseline phase. Toby’s disruptive behavior occurred at high levels during baseline, initially showing an increasing trend before becoming more stable. Toby engaged in low levels of on-task behavior and remained relatively stable during the baseline. During the intervention phase, Toby’s disruptive behavior decreased to an average of 2.6 min (range, 0–6.1 min), while his on-task behavior increased to an average of 25.1 min (range, 22.6–28.3 min). These behavior changes were immediate, with disruptive behavior showing a decreasing trend over time and on-task behavior showing an increasing trend.

Tau-BC coefficients were calculated to estimate the effect size for the two dependent variables (i.e., on-task behavior, disruptive behavior) for each of the three students (i.e., Ryan, Pete, and Toby), which provided an individual effect size for each student’s outcomes for each behavior. In addition, an overall Tau-BC index was calculated for each dependent variable across all three students based on the aggregated data to determine the overall effect size for each behavior. The individual Tau-BC indices were 1.00 (maximum value) for both behaviors. Therefore, the overall Tau-BC coefficient was 1.00, indicating that there was no overlap between the baseline and intervention phases for any student or behavior when pairwise comparisons were made between the phases. This demonstrates a substantial PTR intervention effect on improving student behaviors.

#### 3.1.2. IBRST Data

Figure 2 and Figure 3 display each teacher’s IBRST ratings of student target behaviors in comparison to the IBRST scores converted from direct observations. Overall, the trends and stability patterns evident in the IBRST ratings completed by teachers were similar to those observed in the direct observational data. Although several sessions, particularly for Ryan, had one or two anchor points that deviated from the direct observational data, overall, the results aligned across these two data sources in the baseline and intervention phases. Both Pete and Toby’s teachers consistently rated the student disruptive behavior higher during the baseline than intervention; once they began to implement the intervention, ratings of disruptive behavior decreased by an average of 2–3 rating points. Similarly, Pete and Toby’s teacher ratings for on-task behavior increased by an average of 2–3 anchor points during the intervention phase. It should be noted that there were some overlapping data points between the baseline and intervention for Ryan’s teacher ratings. However, her ratings for disruptive behavior were consistently lower, and the ratings for replacement behavior were consistently higher during the intervention than during the baseline. During the follow-up, teacher ratings showed the maintenance of reductions in disruptive behavior and increases in on-task behavior for Pete, aligning with the direct observational data. However, Ryan’s teacher ratings conflicted somewhat with the observational data, showing a slightly decreased level of disruptive behavior and increased on-task behavior compared to the intervention levels.

#### 3.1.3. Generalization

Generalization probe data indicated that all three students demonstrated substantial improvements in their target behaviors during non-targeted academic time periods when their teachers used the PTR intervention strategies without receiving training and with only one initial feedback session with the researcher. This consistency further demonstrates the generalization of the intervention effects beyond the specific target academic subject times. 

For Ryan, one generalization probe was conducted in the baseline during the generalization period (i.e., reading), where he engaged in disruptive behavior for 11.9 min and on-task behavior for 17.8 min. Three generalization probes were conducted during the intervention phase, with Ryan’s disruptive behavior decreasing to an average of 0.9 min (range, 0.4–1.2 min) and his on-task behavior increasing to an average of 28.9 min (range, 28.2–28.9 min). For Pete, one generalization probe was conducted in the baseline, and four generalization probes were conducted in the intervention phase during the non-targeted writing time. The data showed that his disruptive behavior decreased from 18.9 min in the baseline to an average of 2.5 min (range, 1.6–3.6 min) during the intervention phase, and his on-task behavior increased from 7.7 min in the baseline to an average of 28 min (range, 26.9–28.3 min) during the intervention phase. For Toby, four generalization probes were conducted in the baseline, and three generalization probes were conducted in the intervention during the non-target science period. As with Ryan and Pete, Toby’s disruptive behavior decreased from an average of 25.8 min (range, 24.1–28.9 min) in the baseline to an average of 3.8 min (range, 1.8–7.1 min) during the intervention, while his on-task behavior increased from an average of 2.7 min (range, 1.03–3.6 min) in the baseline to an average of 24.8 min (range, 21.2–26.2 min) in the intervention.

One or two follow-up observations were conducted for Ryan and Pete during the generalization academic time periods. Both students demonstrated maintained improvement in their behaviors, with levels similar to those observed during the intervention phase. Ryan’s disruptive behavior occurred for an average of 3.3 min (range, 2.4–4.2 min), while his on-task behavior was an average of 25.5 min (range 25.2–25.8 min). Pete engaged in disruptive behavior for 3.4 min and on-task behavior for 5.3 min. The maintained behavior improvement during the follow up observations indicates that the teachers continued applying the PTR intervention strategies over time without requiring researcher involvement, and treatment effects were maintained.

### 3.2. Social Validity

Social validity survey data obtained from all four teacher(s) indicated that the PTR intervention was acceptable. The overall average rating for the PTR intervention was 4.2. Except for one question related to the potential permanent improvements in the student’s behavior rated by one of Ryan’s teachers, which was rated as 1, all responses ranged from 3 to 5. These results indicated that the teachers found the PTR intervention goal, procedures, and outcomes to be acceptable. The teachers felt that the goals of the intervention fit with the team’s goals for improvement of the student’s behavior (M = 4.8), and they were willing to carry out the intervention (M = 4.8) and continue to implement after the termination of the study (M = 4.0). The teachers also indicated that the PTR intervention was effective in teaching the students appropriate behavior, and the IBRST was acceptable (M = 4.5) and took little time to complete (M = 4.0). Ryan completed a social validity measure, and his scores indicated that the PTR intervention was acceptable. His mean rating was 4 (range, 3–5). He strongly agreed that the plan helped him meet his school goals and agreed that he was more motivated to participate in class activities than before.

## 4. Discussion

This study had three aims. The first was to evaluate the impact of the PTR intervention on the behaviors of elementary students at risk of EBD and whether teachers would implement PTR intervention with high fidelity. The second aim was to examine the extent to which teachers would implement the PTR intervention plan with fidelity in a non-targeted academic routine while maintaining child behavior improvement. The third was to evaluate teacher and student perceptions of PTR’s social validity. The results indicated that the PTR intervention was implemented with fidelity by teachers, and improvements in the target behaviors for all three student participants were observed. The follow-up generalization probes, in which two of the three teachers implemented the PTR intervention plan in another academic routine with minimal research coach assistance, showed emerging promise that interventions could be generalized across routines and student behavior change can be maintained. Finally, all three teachers and one student found the PTR process and outcomes to be acceptable, providing further support for the implementation of standardized FBA/BIP processes. 

The data from this study support the findings that the PTR process is effective in reducing disruptive behavior and increasing academic engagement or on-task behavior in a school setting [24,25,26,27]. Most importantly, all three teachers implemented the interventions at 80% or higher fidelity during targeted academic time periods. When considering behavioral interventions, it is imperative that teachers are willing to implement strategies. PTR enhances teacher buy-in to the intervention plan by building consensual decision processes and considering teacher preference and context when selecting and developing strategies. Rather than having an expert recommend a large number of strategies for implementation, PTR instead has a coach who guides the teacher to choose one prevent, one teach, and two reinforce strategies that align with the behavioral hypothesis and are preferred by the teacher. After selecting strategies, instead of assuming that the teacher understands the intervention and can implement it as intended, the coach and teacher work together to task analyze each strategy selected. 

This task analysis ensures that the teacher can perform the steps by considering what the teacher thinks is feasible for their situation. Furthermore, the task analysis can then be used to train the teacher to perform the strategy and measure implementation fidelity. Future researchers may want to examine specific components of PTR that increase teacher willingness. This could be the precision of the task analysis or the ability to make choices about the interventions that are included in the plan. Another indicator of teacher willingness is the observation of the use of strategies in other situations without coach guidance. For example, Ryan’s teacher was observed using prevention (antecedent) strategies more often with novel students. An increase in the use of behavior-specific praise with her class was also noted. To monitor student progress and response to the PTR intervention, the participating teachers completed the IBRST on a daily basis. The teachers were involved in developing this monitoring tool as part of the PTR process. In the social validity assessment, they reported that the IBRST required little time to complete and was acceptable for tracking student progress. However, further research is warranted to more fully examine the usability and feasibility of teachers utilizing the IBRST to monitor student progress in their classrooms over time. 

Student willingness and engagement in the interventions is another indicator of effectiveness. All three students expressed excitement about their plans during student training, particularly related to earning points and reinforcement. The PTR process has tools and procedures for obtaining student input on all steps, including the selection of interventions. Deenihan and colleagues [35] described various levels of student involvement in their examination of PTR with autistic high school students, which may have contributed to the success of the plans. Future researchers could deeply explore student involvement in the PTR process that can help educators decide when, how, and to what extent students should be involved. Future researchers might also want to determine which intervention plan components are most effective. For example, Toby’s teacher highlighted the choice strategy as having the greatest impact on his behavioral change. Plan component analysis can contribute toward greater buy-in by reducing the number of strategies to be included, making it more efficient and feasible for teachers. 

The study also extended the findings of Kulikowski et al. [29] by demonstrating the generalization effects of the PTR intervention implemented independently by teachers across students during non-target academic activities. In contrast to the study by Kulikowski et al., which evaluated teacher generalization of the PTR plan for a new child in a community preschool setting, this study focused on teacher generalization of the intervention to non-target academic time periods and its collateral effects on student behavior changes for two of the three elementary students. Without generalization training but with initial feedback, the teachers implemented the intervention successfully, and positive impacts on student behaviors were consistent. This consistency further demonstrates the generalization of the intervention effects. Although follow-up data were limited, Ryan and Pete’s maintained behavior changes multiple weeks after the end of the intervention suggest lasting impacts on their classroom behaviors. The generalized improvements in their disruptive and on-task behaviors suggest that the PTR intervention addresses underlying issues that manifest across situations rather than just behaviors in one setting. 

These findings add to the emerging research of implementing interventions in new routines or activities. In practical applications, the purpose of designing and implementing behavior intervention is to maintain long-term behavior change without the need for interventions [36]. This is typically accomplished by implementing the intervention across times (e.g., follow-phase with probes) and across new routines or people or by fading intervention components until the plan is withdrawn. Unfortunately, the current research related to generalization, and specifically maintenance, is limited. Pennington et al. 2019 [37] reviewed 23 studies of FBAs and BIPs in school settings and found that only 51% examined intervention implementation in a generalization phase. Furthermore, those that did build on the generalization phase were inconsistent in defining generalization, interchanging the term with maintenance. Finally, most of the studies only explored generalization across times and routines, with few examining whether the behavior change was maintained in absence of the intervention. Although the current study adds to the generalization research, it does not allow us to evaluate the maintenance of improved behavior without intervention. Future research is needed that builds on the careful fading of intervention components and withdrawal of the intervention over time to determine if behavior changes are maintained at the same level when the full intervention is removed. This would better assess the real-world sustainability of the behavior improvements and the degree to which the intervention produces lasting changes.

Both of Ryan’s teachers found PTR to be socially valid. They gave the highest ratings to the plan’s acceptability and feasibility of implementing it without it being intrusive or causing discomfort for the student. They liked the strategies, which fit into their current routines. They both gave high ratings for using the IBRST as their progress monitoring tool. However, they both gave relatively lower ratings (i.e., 3) to items that focused on their perceptions that the plan would result in maintaining Ryan’s behavior change, and although they implied that the plan was acceptable and fit into their current routines, they gave lower ratings to the item that asked about the time needed each day to implement the behavior plan. Ryan highly agreed (5) that the plan helped him meet his school goals, and he was more motivated to participate in activities (4). He gave his lowest rating (3) to liking the plan. The high social validity given to PTR may have contributed to teacher buy-in and their high implementation fidelity of the plans. Interventions with higher acceptance can bridge the research-to-practice gap by identifying the intervention components that are more acceptable to teachers and those that need to be modified. Future PTR research should explore adapting intervention components or time necessary to implement intervention plans, planning for the maintenance of behavior change, and examining the impact of the modifications on teacher implementation fidelity, student behavior change, and social validity.

It is important to note that several limitations exist with the current study. The first is the time constraints that occur in typical school settings, which resulted in a delay in conducting baseline data collection for Pete and Toby and collecting a limited amount of intervention data for Toby, which resulted in a limited demonstration of experimental control. Furthermore, the time constraints did not allow the research team to conduct the follow-up evaluation for Toby. Whether it be due to student or teacher absences, testing, or other unplanned events, these factors made consistent data collection via direct observations challenging. Additionally, despite efforts to schedule observations around competing demands, the teachers’ substantial existing responsibilities and limited availability outside instructional hours caused delays in the PTR process. These implementation issues likely contributed to underestimations of generalization and maintenance effects of the intervention. Given that insufficient time and resources are common barriers for schools, future research should investigate methods to streamline and tailor the FBA/BIP process. Addressing logistical barriers could improve fidelity and in turn provide further evidence regarding the maximum potential benefits of interventions. While time constraints affected the process and outcomes of the PTR process, concerted efforts were made to monitor and mitigate limitations. 

Further research with more comprehensive implementation planning and fidelity monitoring procedures is warranted to build on findings from the study. For example, future researchers might consider evaluating the use of the following strategies: (a) intervention implementation planning that includes logistical implementation planning [38]; (b) using visual and auditory reminders and scheduling dedicated time for implementation and data collection in teachers’ daily schedules to prompt and protect intervention activities from competing demands [39]; (c) using technology (e.g., online platforms, video conferencing) to conduct remote coaching and collaboration when in-person meetings are difficult [40]; and (d) advocating for increased priority and resource allocation for the implementation of evidence-based interventions from school administrators [41]. It may also be beneficial for the field to reconsider stringent research publication requirements, such as the preference for concurrent MLB designs over non-concurrent designs and conducting follow-up evaluations. Adjusting these requirements could allow for more flexibility and timely implementation of interventions by teachers in applied settings. 

Another limitation is that, although teachers implemented their plan with a mean greater than 80% and there were improvements in target behaviors across student participants, at times, classroom situations interfered with full implementation. For example, Pete’s teacher would provide behavior-specific praise and dots for on-task behavior but would often miss teaching opportunities and was unable to provide the reinforcement while attending to other students. This inconsistency or inaccuracy in implementing the intervention strategies might have caused variability in his data during intervention. Future research might explore what constitutes adequate fidelity for behavior intervention plan implementation. While this study set the fidelity level at 80%, it is unknown whether that is too high or too low. Furthermore, the quality and dosage of the interventions were not measured, only accuracy of steps. Additional research can explore adding these fidelity measurement components. Another research area that would benefit the field would be to evaluate the specific intervention components that have the strongest impact on student behavior change. By conducting intervention component analyses, time constraints may be addressed by eliminating the components that are unnecessary for effectiveness. Before implementing the PTR process, the existing classroom management systems employed by the teachers focused heavily on punishments for problem behaviors rather than reinforcements for replacement or desired behaviors. This indicates that the teachers likely had more punitive relationships with students engaging in frequent problem behaviors, rather than nurturing bonds. We did not directly measure the quality of student–teacher relationships in the study. Examining how the quality of individual student–teacher relationships impacts the PTR process and outcomes would be an important area for future research.

A third limitation is the extent to which the findings of this study can be generalized to other settings due to the small *N*. This study used a multiple-baseline design across three participants and found similar results for each case, which enhances its generalizability; however, the research should be replicated in future studies to build stronger evidence of the functional relationship between the intervention and the outcomes. In a systematic and meta analytic review of 10 single-case design studies on school-based PTR intervention, Russo et al. [42] used the percentage of goal obtained (PoGo) [43] and Tau-BC [33] to examine the magnitude of the effects of the PTR interventions. The results showed large effects (Tau-BC) and medium to large effects (PoGo) for both concerning behavior and replacement behavior across student participants. The effect sizes were higher for preschoolers and secondary students than for K–5 students and for children with disabilities than for children without disabilities. Thus, empirical support for PTR is emerging. Adding future single-case designs to the meta-analysis would provide more confidence in PTR’s external validity applied to different individuals and settings.

In conclusion, the current study contributes to the growing body of PTR research by implementing it with students with a greater chance of EBD and by generalizing the strategies to a novel academic routine. The results replicate outcomes reported in previous studies, affirming that PTR is an effective FBA/BIP model that can be successfully applied in school settings.

## Figures and Tables

**Figure 1 behavsci-14-00093-f001:**
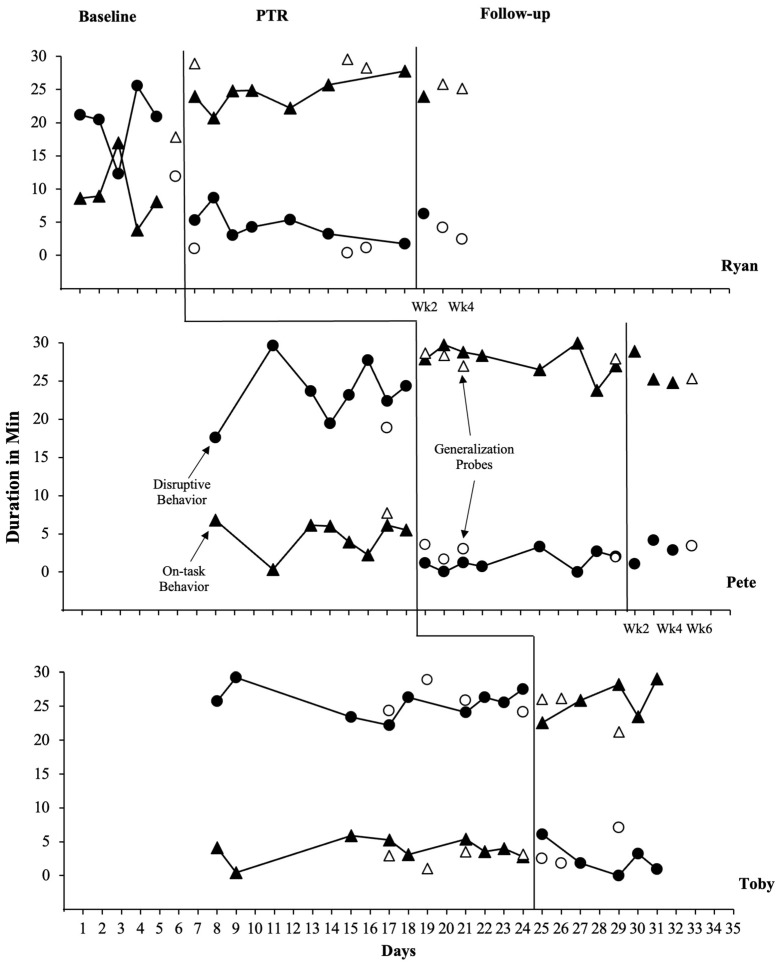
Direct observational data (duration in min) of target behaviors across students and phases.

**Figure 2 behavsci-14-00093-f002:**
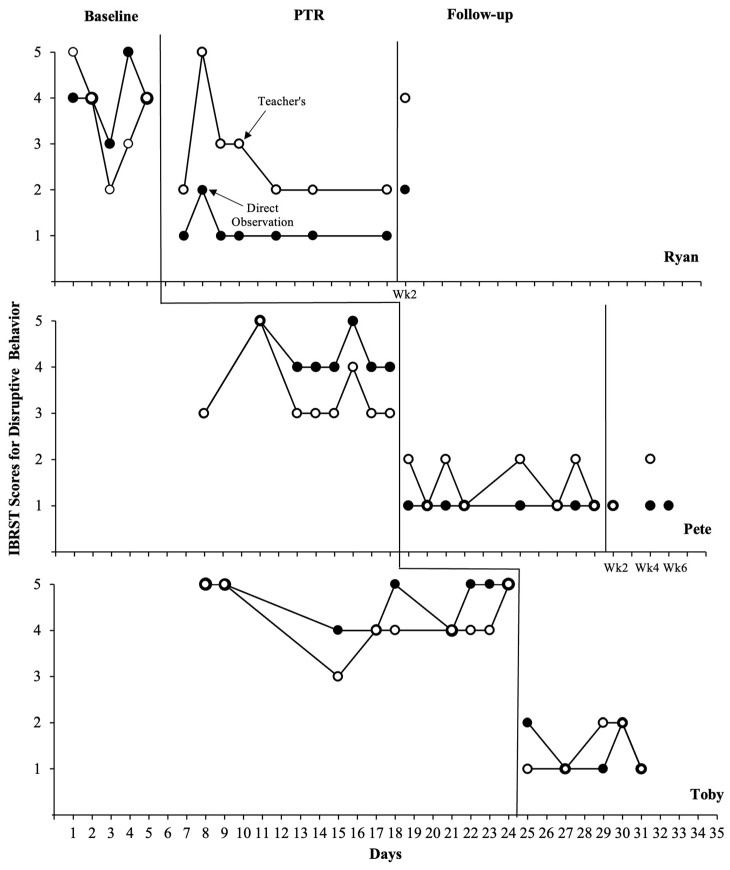
Teachers’ IBRST scores and researcher’s direct observational data (duration in min) converted to IBRST scores for disruptive behavior across students and phases.

**Figure 3 behavsci-14-00093-f003:**
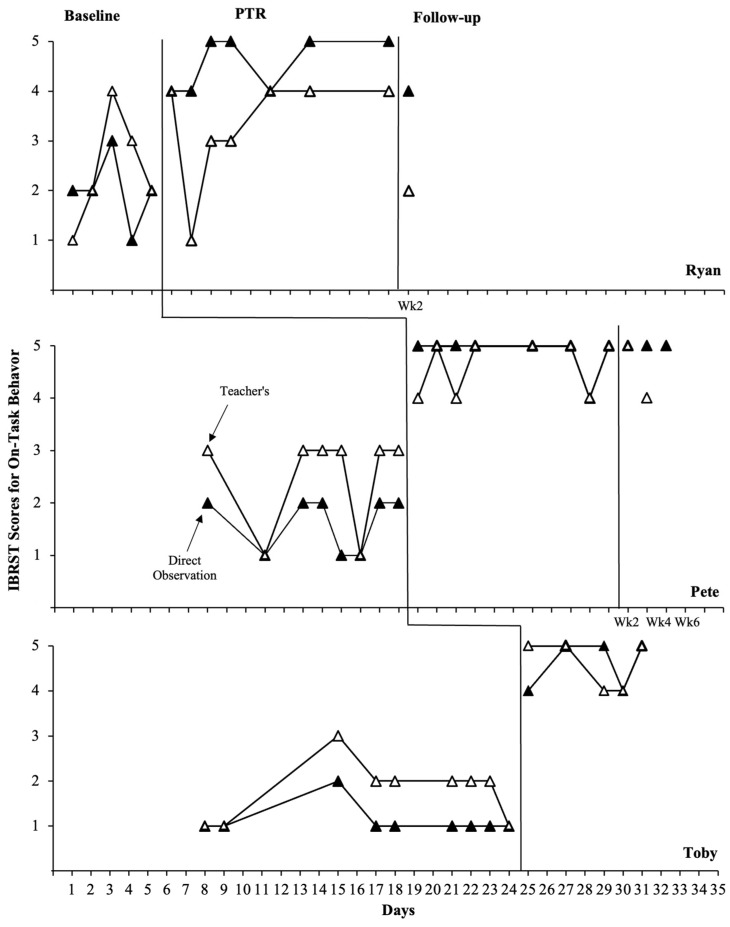
Teachers’ IBRST scores and direct observational data (duration in min) converted to IBRST scores for on-task behavior across participants and phases.

**Table 1 behavsci-14-00093-t001:** PTR strategies selected for each student’s PTR intervention plan.

Student	Prevent Strategies	Teach Strategies	Reinforce Strategies
Ryan	Provide choices (between activities, what materials to use, or where to sit). Provide clear instructions and use visual cues for transition.	On-task behavior: Provide instruction after Ryan is attending, review expectations, redirect avoidance behavior, and briefly discuss alternative behaviors. Problem-solving situations: Check-in with Ryan, provide a reminder to request for help, and provide a reinforcer for skill demonstration.	On-task behavior: Provide points for on-task behavior and choice of preferred activity when the on-task goal is met. Avoidance behavior: Minimize attention, redirect to on-task behavior by using gestures, and ensure task completion.
Pete	Provide choices of tasks. Review choice board with Pete. Provide reminders and cues for on-task behavior.	On-task behavior: Provide instruction after Pete is attending, review expectations, and check-in on understanding. General coping strategies: Briefly discuss disruptive and alternative behavior and provide a reinforcer for skill demonstration.	On-task behavior: Provide behavior-specific praise/points and a choice of preferred activity when the on-task goal is met. Provide a choice of special reward if the goal is met twice in one day.
Toby	Provide choices of tasks. Review classroom rules and use prompts to supplement the rules.	On-task behavior: Provide instruction after gaining Toby’s attention, review expectations for each activity, and check-in on understanding. Social skills: Prompt Toby to engage in desired social skills, briefly discuss alternative behavior, and provide a reinforcer for skill demonstration.	On-task behavior: Provide behavior-specific praise/points and choice of preferred activity when the on-task goal is met. Ensure task completion, even in the presence of challenging behaviors.

## Data Availability

Data are available upon reasonable request from the corresponding author.

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
