# Peer review of "Implementation of the Prevent-Teach-Reinforce Model for Elementary School Students Needing Intensive Behavior Intervention"

_behavsci, 2024, doi:10.3390/bs14020093_

Round 1

Reviewer 1 Report

Comments and Suggestions for Authors

This article has an interesting theme that brings the topic forward. It is well structured and has an in-depth description of the methods, participants and research setting. In addition, solid results are presented in line with the general development of the work. However, there are certain aspects that could be improved, as detailed below.

The concept of disability used at the beginning of the paper is not clear. In the abstract as well as in the introduction, they talk about whether the participants have a disability or not. In the specific case of the sample of this paper, none of the three students has a disability. I do not understand, therefore, why the abstract states "with or without disability". It does not make sense if none of them subsequently has a disability. I understand that in other research on the subject this is relevant, and that, in your review of the literature prior to developing the work, it could be relevant, but not in the specific research where none of them has a disability. You should review this aspect. In addition, you do not clarify whether ED is a disability or not, and if so, what it refers to (lines 31-34).

To finish with this aspect, in case you want to emphasise that this proposal can be worked with students with disabilities, you should clarify everything related to the implementation of PTR with people with disabilities: in what type of disabilities is it implemented? Is there research that shows differences in implementation for students with and without disabilities? Does that research talk about disability in the same terms as you do?

On the other hand, it would be interesting to know what kind of bond exists between individual pupils and teachers, and whether this affects the final results. Behaviour is a multifactorial aspect that should be addressed as such, and not as a single-factor response. 

As for direct observation, it is measured, in some cases, in 5s slots. How can the teacher make a record of such short spaces? To measure the fidelity of the teacher's implementation, the author and the research assistant were present. In addition to the two classroom teachers, there were 4 adults. Did this affect the students' own behaviour?

The results are presented in seconds. However, this is not a standard measure of normative behaviour as far as the subject of the study is concerned. It is a measure more in line with research on the subject, the IBRST data ratings. It gives the impression that a pure quantitative analysis was promoted, discarding the possibility of a mixed analysis, taking advantage of the benefits of quantitative analysis. In this case, more importance could have been given to the teachers' voice, i.e. the Social Validity section. The opinion of teachers who work with students on a daily basis can provide us with data more in line with the concrete reality of the sample, both in terms of development and results. This aspect is also reflected in the conclusions, where only one sentence is devoted to analysing the teachers' opinion.

Finally, two aspects, one minor and one major.

For the minor, correct the percentages of the pupil data in each of the two schools. They do not add up to 100.

The biggest problem of this work is the number of the sample. There are three specific cases, which are difficult to generalize. I understand that this is an investigation that adds to the previous ones, and with it, a corpus is being elaborated that, in the long run, will have enough validity to be generalizable. But the study, by itself, is not. Even so, it is a good case study, which helps to better understand the methodology presented.

Reviewer 2 Report

Comments and Suggestions for Authors

This article, titled "Implementation of the Prevent-Teach-Reinforce Model for Elementary School Students Needing Intensive Behavior Intervention," presents a study evaluating the effectiveness of the Prevent-Teach-Reinforce (PTR) model on elementary school students displaying challenging behaviors. The study involved three students, some with disabilities, and their classroom teachers across two public schools. The PTR process included team collaboration, goal setting, behavior assessment, intervention, and evaluation. The research utilized a multiple baseline across participants design and observed student behaviors during target and general academic periods.

introduction

The text effectively establishes the importance of FBAs and BIPs in addressing problematic behaviors and stresses the significance of understanding the functions of behavior for effective intervention. It rightly identifies the shortcomings in current school systems regarding the adequacy of FBAs and BIPs, emphasizing the need for more effective and standardized interventions.

Additionally, the introduction effectively introduces the Prevent-Teach-Reinforce (PTR) model as a potential solution. It outlines the key components of the PTR model and references several studies supporting its effectiveness, showcasing its positive outcomes across diverse student populations.

However, there are areas for potential improvement. The transition between some paragraphs could be smoother to enhance the flow of information. Furthermore, while the introduction thoroughly discusses the effectiveness of the PTR model in various contexts, it might benefit from a more structured presentation of the existing gaps in research. For instance, clearly delineating the limitations of prior studies on PTR, such as the lack of data on generalization effects across untrained routines or activities, could provide a more coherent understanding of the existing gaps in the literature.

This text describes the method used in a scientific study aimed at implementing a PTR (Positive Behavioral Interventions and Supports, PBIS) intervention in educational settings. The study involves observations and interventions for students showing disruptive behavior.

The method appears comprehensive and detailed, providing a clear overview of the study's setting, participants, inclusion criteria, and behavioral observations. However, there are certain areas where improvements could be made:

·         Clarity and Conciseness: The method section is highly detailed, which is valuable in scientific research. However, some sections could benefit from greater conciseness without losing essential information. For instance, the description of teachers' characteristics and classroom management systems might be streamlined.

·         Behavioral Definitions: The operational definitions of behaviors for observation lack explicit clarity on the metrics used, like how disruptive behavior is quantified. Providing specific details on the metrics would enhance the method's reproducibility.

The Results section presents a detailed analysis of the intervention's impact on three students' behaviors through direct observation and IBRST data. The presentation of findings is clear and follows a structured format, facilitating understanding of the behavior changes observed in each student.

Strengths:

·         Clear Presentation: The section effectively uses figures and textual descriptions to illustrate behavior changes across baseline, intervention, and follow-up phases for each student, providing a clear snapshot of their progress.

·         Detailed Behavior Analysis: The detailed breakdown of each student's behavior, including disruptive behavior and on-task behavior, during baseline, intervention, and generalization probes, offers a comprehensive understanding of the intervention's impact.

Suggestions for Improvement:

·         Interpretation of Conflicting Data: Inconsistencies between direct observational data and teacher ratings, especially for Ryan, need further discussion or potential explanations. Addressing these discrepancies would strengthen the credibility of the findings.

·         Generalization Discussion: While generalization probe data are provided, a deeper discussion on the implications of these findings for the intervention's effectiveness in real-world scenarios or different academic settings would add depth to the interpretation of results.

·         Social Validity Interpretation: Although social validity survey results are presented, further analysis or discussion regarding the specific aspects of the intervention considered acceptable by teachers and students would enhance the section's depth.

The Discussion section offers a comprehensive overview of the study's objectives, outcomes, and implications, showcasing the effectiveness of the PTR intervention in addressing behavioral concerns among elementary-aged students at risk of emotional and behavioral disorders (EBD). The section demonstrates several strengths in interpreting the study's findings and highlighting areas for further investigation:

Strengths:

·         Objective Reflection: The discussion provides a concise summary of the study's aims and outcomes, reflecting on the effectiveness of the PTR intervention in improving targeted behaviors across all three student participants.

·         Comparison and Extension of Previous Studies: It effectively compares the current findings with prior research, especially emphasizing the extension of previous results by demonstrating generalization effects across non-targeted academic activities for multiple students. This contextualization strengthens the significance of the present study's contributions.

·         Teacher and Student Acceptability: The section adequately explores teacher and student acceptance of the PTR intervention, acknowledging their positive responses, which fortify the potential for widespread implementation.

Suggestions for Improvement:

·         Statistical Support: While the section provides a detailed narrative of the observed changes and generalization effects, it lacks statistical analysis to support the significance of the findings. Incorporating statistical measures (e.g., effect size, significance tests) could bolster the validity of the reported effects.

·         Deeper Exploration of Limitations: The limitations section is informative but could benefit from a deeper exploration of the impact these limitations might have had on the study's outcomes. For instance, the discussion about classroom situations interfering with full implementation could be expanded to assess the potential influence on data variability and interpretations.

·         Further Investigation: While the discussion touches upon aspects like teacher willingness, student engagement, and plan components, it would benefit from specific suggestions for future research directions. For instance, proposing studies that delve deeper into the impact of student involvement in the PTR process or investigating the effectiveness of individual plan components would enrich the discussion.

·         Addressing Implementation Challenges: While acknowledging time constraints and classroom situations affecting implementation, the discussion could suggest strategies or recommendations to overcome these challenges for better real-world applicability.

Round 2

Reviewer 2 Report

Comments and Suggestions for Authors

The review comments have been made in detail. Thank you for the work done. Congratulations. 

Author Response

We greatly appreciate you taking the time to carefully go through the changes and provide such positive feedback. Thank you again for your time and helpful evaluations throughout this process.